# The Role of Oxidative Stress in Tumorigenesis and Progression

**DOI:** 10.3390/cells13050441

**Published:** 2024-03-02

**Authors:** Kexin Li, Zhangyuzi Deng, Chunran Lei, Xiaoqing Ding, Jing Li, Changshan Wang

**Affiliations:** Laboratory of Reproductive Regulation & Breeding of Grassland Livestock, School of Life Science, Inner Mongolia University, 49 Xilingol South Road, Yu Quan District, Hohhot 010020, China; kexinli@mail.imu.edu.cn (K.L.); zhangyuzideng@mail.imu.edu.cn (Z.D.); chunranlei@mail.imu.edu.cn (C.L.); xiaoqingding1218@126.com (X.D.); ljdbmonologist@163.com (J.L.)

**Keywords:** oxidative stress, tumorigenesis, mitochondria, cell aging, cell death, tumor treatment

## Abstract

Oxidative stress refers to the imbalance between the production of reactive oxygen species (ROS) and the endogenous antioxidant defense system. Its involvement in cell senescence, apoptosis, and series diseases has been demonstrated. Advances in carcinogenic research have revealed oxidative stress as a pivotal pathophysiological pathway in tumorigenesis and to be involved in lung cancer, glioma, hepatocellular carcinoma, leukemia, and so on. This review combs the effects of oxidative stress on tumorigenesis on each phase and cell fate determination, and three features are discussed. Oxidative stress takes part in the processes ranging from tumorigenesis to tumor death via series pathways and processes like mitochondrial stress, endoplasmic reticulum stress, and ferroptosis. It can affect cell fate by engaging in the complex relationships between senescence, death, and cancer. The influence of oxidative stress on tumorigenesis and progression is a multi-stage interlaced process that includes two aspects of promotion and inhibition, with mitochondria as the core of regulation. A deeper and more comprehensive understanding of the effects of oxidative stress on tumorigenesis is conducive to exploring more tumor therapies.

## 1. Introduction

Recently, the relationship between oxidative stress and cancer has become a hot issue. Oxidative stress is closely related to tumor development and can interfere with tumor cell fate through a complex regulatory network. Considering that a major challenge in cancer treatment lies in the diverse mechanisms of tumor development and evasion, targeting oxidative stress may be a comprehensive approach compared with some single efficacy targets. ROS, as one of the normal cellular metabolites, exists in various forms, such as radicals possessing a single unpaired reactive electron in the outermost orbital, including superoxide anion (O_2_^•−^), hydroxyl radical (OH^•^), carbonate radical anion (CO_3_^•−^), nitrogen dioxide (NO_2_^•^), alkoxyl/alkyl peroxyl (RO^•^/ROO^•^), etc., and non-radicals lacking unpaired electrons and characterized as two-electron oxidants, including hydrogen peroxide (H_2_O_2_), nitric oxide (NO), hypochlorous acid (HOCl), etc. [1,2]. With an intensive oxidative capacity, ROS can attack nucleic acids, proteins, and lipids, resulting in DNA damage, lipid peroxidation [3], and altering protein post-translational modification, represented by redox modification [4] and phosphorylation [5]. It is known that ROS originates from two main pathways: an endogenous pathway with mitochondria as the primary source, containing the endoplasmic reticulum, NADPH hydrogenase, and catalase, among which NOXs are considered as the central enzyme family for ROS production [6]; and an exogenous pathway including external factors such as radiation, chemotherapy, inflammatory factors, and air pollution [7]. The basic relationship between oxidative stress and cancer has been clearly demonstrated. Relatively high ROS levels can induce DNA mutations and pro-oncogenic signaling pathways to promote tumor formation, while excessive ROS levels can induce tumor cell death [8]. This implies that early tumor formation can be prevented by removing relatively high levels of ROS, or cancer cells can be killed explicitly by promoting the production of excessive levels of ROS in cancer cells. In this review, we discuss the crucial role of ROS-based oxidative stress in various aspects of tumorigenesis and progression. Based on its features and cell fate determination, some potential treatments are proposed.

## 2. Oxidative Stress and Tumorigenesis

According to the hallmarks of cancer concluded by Douglas Hanahan and Robert Weinberg in 2000 and 2011, genome instability and mutation are enabling characteristics of cancer [9]. The mutability of cancer is achieved through internal factors such as spontaneous mutation accumulation and external factors such as the environment and radiation [10]. ROS, as cellular intermediates of both factors, will directly attack DNA, triggering various forms of DNA damage, such as DNA strand breaks, which affect the expression levels of key genes such as proto-oncogenes, oncogenes, and DNA damage repair-related genes, and promotes tumorigenesis [11,12]. In addition, it has been elaborated that ROS induces mutagenic break repair and the SOS response via damaging bases in DNA, then pausing the replisome and allowing the critical switch from high fidelity to error-prone DNA polymerases, which provoke more carcinogenic mutation [13,14]. It has been shown that ROS can affect tumorigenesis and transformation by oxidizing cysteine residues, which activates the three most common oncogenic switch genes in human cancers, *HRAS*, *NRAS*, and *KRAS* [15]. Mitochondrial ROS (mtROS) had a critical role in tumorigenesis through the ERK-MAPK signaling pathway in a mouse model of oncogenic Kras-mediated lung cancer [16]. Also, in Kras-driven mouse models of pancreatic cancer, ROS inhibition using NAC and MitoQ was found to significantly reduce the development and progression of precancerous lesions [17].

Epigenetic regulation is another vital mechanism altering expression of tumor-related genes [18]. It was found that ROS, as a catalyst for DNA methylation, is extensively involved in the regulation of aberrant hypermethylation and overall hypomethylation levels in the promoter region of tumor suppressor genes (TSG) such as *CDX1* through upregulation of DNA methyltransferase (DNMT) expression or the formation of DNMT-containing complexes [19,20]. Simultaneously, ROS directly induced LINE-1 hypomethylation and *RUNX3* promoter hypermethylation in bladder cancer cell lines, uroepithelial cell carcinogenesis [21], and hepatocellular carcinoma (HCC), making ROS-induced *RUNX3* hypermethylation promising as a practical and valuable biomarker for diagnosis [22,23] (Figure 1).

## 3. Oxidative Stress and Tumor Metabolism

Tumor cell metabolic reprogramming is a crucial pathway in ROS-induced cancer cell development, and its primary role is to maintain tumor cell adaptive capacity [24]. As early as the 1920s, the German biochemist Warburg pointed out that tumor cells had higher ROS levels than normal cells because of increased aerobic glycolysis (Warburg effect) in cancer cells [8]. The inhibition of glycolysis directly leads to the death of colon tumorigenic cells (HCT116) and lymphoma cells (Raji) [25]. In addition to a preference for aerobic glycolysis and higher levels of ROS, tumor cell metabolic reprogramming is also characterized by enhanced lipid synthesis, abnormal amino acid metabolism, increased lactic acid production, and alteration of the antioxidant system. Cancer cells also possess more powerful antioxidant defenses in contrast to its multitudes of ROS [26]. Intensive ROS scavenging, including peroxiredoxin 1 (PRDX1) [27], SOD2 [28], CAT, GSH-PXs, thioredoxins (TRXs), and GSH, can be upregulated by the activation of *TNFα*, *Nrf2*, *HIF1α*, *AMPK*, and *PGC1α*, protecting cancer cells from damage and subsequent cell death [29,30]. Furthermore, NADPH, which is perceived as a pivot in the antioxidant system, with the capacity of renewing reduced glutathione (GSH) and thioredoxin (TRX), is drastically produced in cancer cells via fostering the pentose phosphate pathway, malic enzymes, one-carbon metabolism, etc. [31]. As research continues, mutual reinforcing between metabolic reprogramming and oxidative stress in cancer cells is gradually being uncovered. It has been suggested that PML, as an ROS sensor, is located in both the nucleus and MAM where it regulates the Warburg effect and metabolic reprogramming related to the division, differentiation, and chemical sensitivity of cancer cells [32,33]. Gemcitabine-induced ROS activates KRAS/AMPK signaling, inducing metabolic reprogramming, and enhances stem cell-like properties in pancreatic cancer [34]. Furthermore, sirtuin, including the SIRT1-SIRT3 axis, a kind of NAD^+^-consuming enzyme regarded as a stress responder, has been demonstrated to trigger metabolic reconstitution and affects the ROS level by deacetylating and activating metabolic enzymes and signaling molecules such as FOXO3a, PGC-1α, TFAM, Drp1, mTOR, and PINK1/Parkin [35,36,37]. In addition, ROS can activate NF-κB [38], NRF2 [38,39], and KHK-A [40] to reduce ROS production. Furthermore, ROS can directly regulate the function of metabolic enzymes through redox modification, which has been demonstrated in key redox-sensitive residues such as cysteine oxidation/S-sulfenylation/S-glutathionylation/S-nitrosylation and tyrosine nitration [4]. For example, oxidative stress contributes to pancreatic ductal adenocarcinoma via inhibiting the arginine methylation of malate dehydrogenase 1 (MDH1) [41].

## 4. Oxidative Stress and Tumor Cell Proliferation

The most important feature of tumor cells is uncontrolled proliferation and growth, and oxidative stress based on ROS levels will directly affect their growth state [42]. That is, the ability of tumor cells to adapt to ROS is crucial for their proliferation [43]. On the one hand, ROS plays an important role in promoting the regulation of tumor proliferation. It was found that the survival rate of pancreatic cancer cells significantly increased after stimulation with high levels of ROS triggered by 5-lipoxygenase (5-LO) and NADPH oxidase 4 (Nox4) [44]. Also, oxidative stress-mediated mitogen-activated protein kinase phosphatase (MKP-3) deficiency is significantly correlated with enhanced tumorigenicity in ovarian cancer cells [45]. In contrast, ROS, as a secondary messenger molecule, can directly mediate the activation of PDGF [46], EGF [47], and MAPK [46] or lead to the inactivation of PTEN [48] to participate in the regulation of tumor cell proliferation [49]. For example, copper chaperones of superoxide dismutase promote breast cancer cell proliferation through ROS-mediated MAPK/ERK signaling [28]. Inactivation of PTEN due to H_2_O_2_ over-activates the PI3K/AKT/mTOR signaling pathway to promote breast cancer cell proliferation [50].

On the other hand, ROS can also inhibit the proliferation of tumor cells in several ways. It was found that high levels of ROS mediated AKT-dependent signaling pathways to effectively inhibit the proliferation of rectal cancer (CRC) cells [51]. In addition to this, high levels of ROS can also inhibit tumor growth through sustained activation of cell cycle inhibitory factors. It was noted that significant elevation of p27 ^KIP1^ leads to quiescent cell cycle arrest in the G_0_ phase [52], whereas silencing p27 ^KIP1^ significantly reverses this arrest and re-enters the cell cycle [53]. By contrast, the most frequently reported are p21 ^WAF1/CIP1^ (CDKN1A) and p16 ^INK4A^ (CDKN2A), the accumulation of which first leads to the hypophosphorylation of retinoblastoma protein (RB) and then inhibits the trans-activation of *E2F* genes involved in nucleotide metabolism and DNA synthesis [54], leading to termination of the regular cell cycle and ultimately inhibiting cell proliferation [55] (Figure 2).

## 5. Oxidative Stress and Tumor Immunity

Oxidative stress exerts a dual effect on tumor immunity. Oxidative stress not only plays a pivotal role in antitumor immune cell differentiation, maturation, and activation, but also imposes an inciting effect on tumor immune escape via evoking tumor-associated immune cells and wreaking havoc on the antitumor system.

Firstly, a variety of immune cell-derived exosomes, represented by macrophage-derived exosomes, implement a tumoricidal effect by directly releasing ROS [56]. Furthermore, ROS indirectly induces an antitumor effect by boosting immune cells. For example, PRAK deficiency-induced ROS accumulation impairs the differentiation of Th17 cells and antitumor immunity though disrupting phosphorylation of STAT3 [57]. ROS spurs dendritic cell maturation by activating p38-MAPK and ERK1/2 [58]. In addition, ROS is involved in the activation of many immune cells, like CD8+ T cells [59,60], macrophages^TG^ [61], and dendritic cells [62]. NLRP3 inflammasome is a crucial component of the innate immune system mediated by ROS [63] and contributes to various cancers like gastrointestinal tract [64,65] and breast cancer [66], while it is also demonstrated to be tumor-promoting in gastric and skin cancer [67]. The cGAS/STING pathway, which has been reported to be ROS mediated, exerts a crucial dichotomous function in the antitumor process, including agitating type I IFN induction in DCs, prompting antitumor CD8+ T cell responses, maintaining CD8+ T cell stemness, etc. [62,68,69,70]. While in some tumor cells, enriched cGAS/STING signaling kindles a tumor-promoting function by activating NF-κB, TBK1, and IRF3 [71]. In addition, the aforementioned ROS-associated metabolic reprogramming also occurs in T cells, prompting CD4+T memory cell and CD8+T memory cell formation and survival [72,73].

Cancer development and immune escape cannot be separated from the interaction between tumor cells and the tumor microenvironment (TME), and the involvement of oxidative stress is essential in this process [74]. On the one hand, ROS takes part in activating and inhibiting the function of immune cell such as myeloid-derived suppressor cells (MDSCs) [75], regulatory T cells (Tregs) [76], and tumor-associated macrophages (TAMs) [77]. For example, SUMO-specific protease 3 (SENP3) accumulation triggered by ROS is involved in deubiquitinating modification of transcription factor BACH2 and its activity in maintaining Treg cell-mediated tumor immunosuppression [78]. In malignant melanoma, mitochondrial ROS produced by TAMs stimulates MAPK/ERK activity, which leads to the secretion of TNF-α and promotes tumor cell invasion [77]. On the other hand, the ROS-induced tumor-promoting microenvironment inhibits tumor-killing cells, such as cytotoxic T lymphocyte (CTL) [79] and CD8+ tumor-infiltrating lymphocytes [80]. For example, tumor-associated neutrophils produce O_2_^•^ mediated by NOX2, inhibiting the expansion of γδ T cells, which promote tumor development by producing IL-17 [81]. Recent studies have indicated that the PRAK-NRF2 axis, which is associated with ROS, is essential for Th17 cells to maintain antitumor effects [57]. Phosphatase PAC1, as an oxidative stress responder [82] and a negative regulator of the immune system, specifically inhibits T lymphocyte defense and promotes tumor immune escape [83]. In addition, the potential of converting M2-TAMs into the immune-promoting M1 subtype has been identified as a promising approach to combat clinically challenging carcinomas [84]. It is ROS in the TME that promotes the polarization of M2 tumor-associated macrophages (TAMs) [85]. Furthermore, T cells in a cellular stress response state (TSTR) are predominantly observed in the TME, which contributes to immunotherapy resistance [86] (Figure 3).

## 6. Oxidative Stress and Tumor Metastasis

Increasing evidence suggests that higher levels of ROS are critical for promoting and maintaining malignant biological behaviors of cancer cells, such as their aggressive metastatic phenotype [87]. In order to achieve the malignant transformation of tumors, early-stage tumor cells usually use the epithelial–mesenchymal transition process (EMT) to invade neighboring stromal cells [88,89]. During this transformation process, ROS promotes tumor metastasis by inducing Rho family guanosine triphosphatase-dependent cytoskeletal rearrangements, promoting matrix metalloproteinase-dependent extracellular matrix protein degradation, and accelerating hypoxia-inducible factor-dependent angiogenesis [49]. Deacetylated SOD2 fosters mitochondrial antioxidant properties, thereby protecting cells from oxidative damage and inhibiting tumorigenesis [90]. However, in the process of tumor development, SOD2 is modified by acetylation, which instead increases mtROS to promote hypoxia signaling, thus promoting EMT in breast cancer cells [91]. Similarly, some factors regulated by redox, like HSF1 [92], NF-κB [93], and MMP [94], also promote metastasis. For example, high levels of ROS in tumor cells activate NF-κB, promoting transcription factor (Snail) expression, downregulating epithelial calmodulin (E-cadherin), and promoting the expression of neural calmodulin and waveform protein, which leads to disruption of cell–cell junctions and triggers the EMT process, stimulating tumor cell metastasis [95]. Furthermore, the downregulation of carnitine palmitoyltransferase 2 (CPT2) induces the ROS/NF-κB pathway in ovarian cancer to promote tumor growth and metastasis [96].

In addition, several studies have shown that ROS can also regulate protein hydrolases matrix metalloproteinases (MMP) [97] and serine proteases, leading to ECM [98] degradation and interfering with the invasive phenotype of tumor cells. For example, G6PD promotes ROS production and activates the MAPK signaling pathway in ccRCC cells, promoting MMP2 overexpression in ccRCC cells and clear cell renal cell carcinoma invasion [94]. In addition, AE-BCT inhibits MMP-9 activity by suppressing ROS-mediated NF-κB activation, thereby significantly reducing the metastatic activity of highly malignant cancer cells [99]. The effect of oxidative stress on EMT was also found to be variable. In stable non-small cell lung cancer cell lines (NSCLC), N-acetylcysteine (NAC) treatment can reduce ROS levels and inhibit EMT phenotypic transformation, which in turn restores the sensitivity of gefitinib-resistant NSCLC cells to gefitinib [100]. Also, in oral squamous cell carcinoma (OSCC) with aberrant expression of serine-threonine protein kinase A (AURKA), the knockdown of AURKA can increase ROS levels and inhibit EMT [101].

## 7. Oxidative Stress and the Relationship Between Aging and Tumors

Cellular senescence, i.e., irreversible arrest of proliferation, is composed of replicative senescence (RS) and stress-induced premature senescence (SIPS). It can lead to cancer development and age-related diseases [102,103]. Senescence was once thought to be the antithesis of tumorigenesis and progression as a universal barrier all tumor cells must overcome [104]. However, in recent years, more and more research has revealed that cellular senescence can also promote hyperplastic pathologies, including cancer [105]. Meanwhile, oxidative stress plays a critical role in the interaction between senescence and cancer, as cellular senescence responds to long-term cellular stress [104].

Oxidative stress directly inducing tumor aging has been well demonstrated. ROS, a product of oxidative stress, plays an important role in stress-induced premature senescence and contributes to the biochemical and molecular changes required for tumor formation, promotion, and progression [106]. For example, ROS triggers cellular DNA double-strand breaks and the associated ATM signaling pathway [107,108,109] or activates senescence-related signaling pathways such as P53/P21 [110], ASK1/JNK/p38 [111], and so on, leading to the inhibition of tumor cell proliferation and activation of antitumor immunity. As one of the hinges of the cellular stress response, mitochondria play a pivotal role in oxidative stress and stress-induced premature senescence. In colorectal cancer cells, artesunate treatment-induced mitochondrial dysfunction can drastically spur mitochondrial ROS generation, thereby promoting cell senescence [112], which has become a vital target of cancer therapy [113].

The role of cellular senescence in tumorigenesis and progression is many things. Senescence drives both aging and tumors, most likely by promoting chronic inflammation and the senescence-associated secretory phenotype (SASP) [105]. The SASP component consists of several chemokines and cytokines that activate immune surveillance and bring about innate and adaptive immune responses to clear senescent and proliferating tumor cells, and enhancing cancer senescence-induced tumor suppressive capacity can support tumor cell growth arrest [114,115,116]. On the contrary, in recent years, an auxo-action for tumorigenesis has been demonstrated in senescence. SASP promotes malignant phenotypes in nearby cells [105] and formats the tumor-permissive microenvironment [117]. In an indirect way, stress-induced immune cell senescence promotes tumor immune escape [83].

Oxidative stress is one of the reasons senescence is induced in SIPS and takes vital part in the processes of aging and cellular senescence. Apart from ROS oxidizing cysteine residues, activating the three most common oncogenic switch genes in humanity cancers [118], ROS possesses the capability to regulate DNMT activity, which is pivotal to the role of aging in tumors [20,119]. Moreover, ROS can activate various signaling pathways associated with cell senescence. For example, ROS-mediated activation of the ASK1 signalosome subsequently activates the p38 MAPK and SAPK/JNK pathways, which promotes senescence by oxidizing Trx [120]. TP53/CDKN1A (p21) [121], pRB/CDKN2A (p16) [122], and other decisive pathways of cell aging are affected by ROS directly or indirectly.

Normally, replicative senescence occurs accompanied by telomere attrition during a series of cell divisions [123]. Then, senescent cells are cleared via apoptosis or phagocytosis mediated by SASP and the innate immune system, such as NK cells [124,125]. However, erosion of telomeres can induce mitochondrial dysfunction and oxidative stress through the p53-PGC-1α-NRF-1 axis [126].

## 8. Oxidative Stress and the Relationship Between Death and Tumors

Excessive ROS levels can induce tumor cell death [8]. There are four primary forms of ROS-induced tumor cell death. The first is the mitochondrial apoptotic pathway. Oxidative stress stimulation directly affects mitochondrial membrane potential by disrupting mitochondrial inner membrane permeability [127], mediating pathways like bax/bcl-2-cyt-c [128] and p38 MAPK, and phosphorylating HSP27 [129], all of which promotes the release of cytochrome c (Cyt c) and activates caspase families, directly driving cancer cell apoptosis [130]. Notably, the mitochondrial permeability transition pore (MPTP) integrates ROS-induced cell apoptosis and calcium signaling through activation by both ROS and Ca^2+^, ensuing an influx of Ca^2+^ and release of Cyt c [131]. The second refers to the endoplasmic reticulum (ER) stress-mediated apoptosis pathway. Research suggests that ROS can cause disorder of Ca^2+^ homeostasis in the endoplasmic reticulum, via damaging RyR and IP3R gating and suppressing Ca^2+^-ATPase activity, thus inducing apoptosis in cancer cells such as prostate cancer cells [132,133]. The third refers to a form of iron-dependent cell death (ferroptosis) caused by the lethal accumulation of lipid ROS, which has recently been identified in various cancers [134,135,136]. Upregulated iron concentrations with elevated levels of ROS stimulate the expression of tumor suppressor p14^ARF^ (CDKN2A) and activate p53, which ultimately inhibits NRF2 activity to promote the onset of ferroptosis [137]. The fourth refers to the p53-mediated apoptotic pathway, which has intricate crosstalk with other pathways. It has been demonstrated that p53 plays a synergistic role with pro-apoptotic factors such as the pro-apoptotic proteins p21^cip1^ and BAX, promoting the mitochondrial apoptotic pathway [138]. In addition, p53 can dually regulate ferroptosis via diverse pathways [139]. Most interesting, it has been suggested that p53 will transfer to the mitochondria to regulate mitochondrial membrane potential and trigger cellular apoptosis under oxidative stress [140,141], and several proteins proven to interact with p53 directly or indirectly also locate in the mitochondria under the state of oxidative stress, such as MDM2 [142], FOXO3a [143,144], ATM, c-Abl [145], Parkin [146], and TERT [144] (Figure 4).

## 9. Oxidative Stress and Tumor Treatment

In theory, tumor therapies can be conducted via contradictory approaches, namely reducing oxidative stress to alleviate tumorigenesis [150], metabolic reprogramming [151], metastasis [152], immune escape [153], and transformation of tumor-promoting cells, such as CAFs [154], or bursting oxidative stress to induce aging and dying of tumor cells and tumor-associated cells. Traditional treatment focuses on directly stimulating ROS though radiotherapy; pro-oxidation chemotherapy drugs, including arsenic trioxide [155] and gemcitabine [156]; and drugs that non-specifically target the mitochondrial electron transport chain, such as atovaquone [157] and rotenone [158]. By contrast, with a capacity to scavenge ROS and interact with oxidative stress-related signaling pathways, polyphenols (such as quercetin [159] and curcumin [160]) are widely used in antitumor therapies to promote antioxidation [161]. Nowadays, with a more in-depth understanding of the mechanism of oxidative stress in tumorigenesis and progression, more and more therapies aim to regulate the metabolic processes of redox substances and target crucial signaling pathways. Regulating metabolic processes can be achieved by drugs that target key redox enzymes system. Drugs targeting GSH and the key enzymes taking part in the production and functioning of glutathione (γ-GCS, GSTs, and xCT) are assessed for their ability to disrupt self-protection responses to oxidative stress and drug efflux [162]. Great significance has been attached to blocking the oxidative stress-associated self-protective signaling pathways such as AMPK, NF-κB, Nrf2, c-Jun, and HIF-1α (Table 1). In addition, the antibody-drug-conjugate (ADC) strategy has become a darling of research and clinical treatmentby specifically and efficiently killing cancer cells. Oxidative stress can provide a promising vision for the linker design owing to its heterogeneous existence in tumor cells and their microenvironment. GSH has been reported to work, and other oxidative stress-relevant targets still need to be discovered [163,164].

In general, according to current clinical research and applications, use of pro-oxidants has developed into a relatively mature antitumor therapy with a long history and diverse roles, including prompting tumor cell senescence, damage, death, and regulating pivotal signaling [8,165]. Meanwhile, pro-oxidants tend to incite various side effects, such as nerve damage and bone marrow suppression [166] and may lead to ROS-induced drug resistance [167,168]. Antioxidants, on the contrary, are reckoned as possessing relatively less side effects when remedying cancer via regulating metabolism and alleviating OS-induced damage [169]. One of the most famous drugs is metformin, which has been demonstrated to reduce mtROS production and inhibit migration and invasion in breast cancer [170].

However, antitumor therapies based on oxidative stress sometimes contradict the expected effect. Inhibitors of EGFR and KRAS, both of which are associated with ROS [34,171], have been widely used in clinical studies with various antitumor effects. For example, adagrasib primarily targets NSCLC by alleviating intratumoral immunosuppression [172], MRTX1133 potently suppresses pancreatic cancer tumor growth via increasing cellular apoptosis and impeding proliferation [173], and lapatinib has been approved for the treatment of breast cancer [174]. While EGFR and KRAS inhibitors have also been demonstrated to slash oxidative stress [175], EGFR-TKI resistance promotes NSCLC by mediating ERRα re-expression, then detoxifying ROS [176]. As a vital member of the antioxidant system, Nrf2 is an ideal target of oxidative damage and cancer therapy. Meanwhile, hyperactivated Nrf2 presents pro-tumorigenic activity and is associated with a worse clinical prognosis [177]. It is also regarded as a marker of the cancer-associated fibroblast (CAFs) phenotype because of inciting the expression of genes characteristic for CAFs in skin fibroblasts, then deteriorating tumor development [178]. Clinical trials like SELECT and ATBC have indicated that tumor therapies based on antioxidative approaches such as dietary supplementation with vitamin E, beta carotene, and so on do not always achieve the expected goals [179,180] and sometimes promote tumor development instead [181]. According to a set of facts like oxidative stress exerting a dichotomous effect on tumor, the intratumoral heterogeneity [182], as well as the oxidative stress tolerance of cancer stem cells and drug-resistant sub-populations [183,184], the intricate interactions with tumor microenvironment members like CAFs, plus the cellular location of ROS where antioxidant elimination may also contribute to the observed failure due to inhomogenous distribution of ROS in the cell [185], we can draw the conclusion that practical effects on tumors are discrepant and largely determined by the multi-dimensional heterogeneity of tumors. On the other hand, excessive antioxidants may jeopardize normal physiological process relying on ROS, like immune killing. In a nutshell, the study of cancer therapy based on oxidative stress should comprehensively take into consideration multiple factors to achieve optimum therapy via precise targeting.

At present, clinical research still faces some problems, such as promoting mutations of some key genes that are undruggable [186] and plenty of first-line drugs target multiple kinases, causing non specificity and side effects. Finding more targets for the oxidative stress responses that protect cancer cells from death is necessary, while drugs specifically targeting key signaling molecules, even one of their downstream pathways, must also be developed. There is a bright prospect for much more precise treatment if drugs are developed to block the formation of a specific complex without affecting their individual functions. This reveals the intense need to understand the specific mechanisms of oxidative stress affecting tumors.

**Table 1 cells-13-00441-t001:** Current clinical applications of antitumor therapies based on oxidative stress. Drug names, properties, applicable cancer types, clinical research phase, and specific mechanisms are explained in detail in the table. Some drugs act on tumor-associated cells or the TME, and they are specifically described in the Function section. The information comes from references and the website ClinicalTrials.gov (https://clinicaltrials.gov/).

Drug Name	Type	Clinical Phase	Specific Mechanism	Function	References
Arsenic trioxide	Chemically synthesized drug	In p53-mutated pediatric cancer, phase 2	Including autophagy, apoptosis, necroptosis, and ferroptosis	Promote oxidative stress in tumor cells	[187]
Gemcitabine	Chemically synthesized drug	In biliary tract cancer, phase 3 trial	Inhibiting nuclear replication, promoting p-STAT3 binding to the promoters of Bmi1, Nanog, and Sox2 genes.	Promote oxidative stress in tumor cells	[156,188,189,190]
Elesclomol	Chemically synthesized drug	In ovarian, fallopian tube or primary peritoneal cancer, phase 2	Promoting cupproposis and killing cancer cells	Promote oxidative stress in tumor cells	[191]
Rotenone	Natural active substance	In colon cancer	Inhibiting the PI3K/AKT/mTOR signaling pathway	Promote oxidative stress in tumor cells	[192]
Fucoidan	Natural active substance	In hepatocellular carcinoma, phase 2	Boosting ROS and mitochondrial superoxide generation and draining ATP	Promote oxidative stress in tumor cells	[193]
2-ME	Chemically synthesized drug	In patients with solid tumors, phase 1	Inhibiting angiogenesisin, increasing CD3+ cell number and promoting tumour necrosis.	Promote oxidative stress in tumor cells	[194]
Naringenin	Natural active substance	In human tongue carcinoma CAL-27 cells	Inducing cell death via modulation of the Bid and Bcl-xl signaling pathways	Promote oxidative stress in tumor cells	[195]
BT-Br	Chemically synthesized drug	In castration-resistant prostate cancer DU145 cells	Binding to NADPH and inducing ferroptosis	Promote oxidative stress in tumor cells	[196]
Atovaquone	Chemically synthesized drug	In non-small cell lung carcinoma, early phase 1	Inducing tumor cell apoptosis by elevating ROS levels	Promote oxidative stress in tumor cells	[197]
Metformin	Chemically synthesized drug	In advanced breast cancer, phase 2	Increasing FOXO3a, p-FOXO3a, AMPK, p-AMPK, and MnSOD levels	Inhibit oxidative stress in tumor cells	[198]
Rapamycin	Chemically synthesized drug	In angiofibromas, phase 2	Targeting mTOR, inhibits tumor proliferation	Inhibit oxidative stress in pre-cancerous cells	[199]
Pirfenidone	Chemically synthesized drug	In neurofibromatosis type 1 and progressive plexiform neurofibromas, phase 2	Suppressing CAF activation	Inhibit oxidative stress in CAF cells	[200]
ME-143	Chemically synthesized drug	In refractory solid tumors, phase 1	Targeting NADPH oxidase, blocking ROS production	Inhibit oxidative stress in tumor cells	[201]
Carboplatin	Chemically synthesized drug	In locally advanced triple negative breast cancer, phase 2	Facilitating early and durable CAR T cell infiltration	Promote oxidative stress in TME	[202]
Apatinib	Chemically synthesized drug	In metastatic colorectal cancer, phase 2	Alleviating hypoxia, increasing infiltration of CD8+ T cells, reducing recruitment of TAMs	Promote oxidative stress in TME	[153,203]
Propofol	Chemically synthesized drug	In pediatric tumor, phase 4	Inducing oxidative stress and apoptosis	Promote oxidative stress in tumor cells	[204]
Doxorubicin	Chemically synthesized drug	In advanced solid tumors, phase 1	Perturbing mitochondrial structure and function in tumor cells	Promote oxidative stress in tumor cells	[205]
Sunitinib	Chemically synthesized drug	In advanced solid tumors, phase 1	Alleviating the tumor hypoxia, improving pericyte coverage on endothelial cells	Promote oxidative stress in TME	[206]
Salidroside	Natural active substance	In human gastric cancer cell line	Downregulating Src-associated signaling pathway and HSP70 expression	Inhibit oxidative stress in tumor cells	[207]
Lipoxin A4	Natural active substance	In pancreatic cancer cells	Suppressing the ROS/ERK/MMPs pathway	Inhibit oxidative stress in tumor cells	[208]
Lobaplatin	Chemically synthesized drug	In human gastric carcinoma cell line BGC-823	Decreasing mitochondrial membrane potential	Promote oxidative stress in tumor cells	[209]
Quercetin	Natural active substance	In metastatic breast cancer, phase 1	Inhibiting signaling pathways, including MAPK/ERK1/2, JAK/STAT, AMPKα1/ASK1/p38, etc. and inducing cell cycle arrest	Inhibit oxidative stress in tumor cells	[159]
Curcumin	Natural active substance	In advanced pancreatic cancer, phase 2	Promoting apoptosis through inhibiting NF-κB	Inhibit oxidative stress in tumor cells	[160]
α-T-K	Chemically synthesized drug	In clinical immunotherapy of sensitized anti-PD-1	Reprogramming M2 macrophages, elevating the curative effect of PD-1 antibody	Inhibit oxidative stress in TME	[85]
Artesunate	Natural active substance	In hepatocellular carcinoma, phase 1	Promoting the accumulation of intracellular lipid peroxides to induce cancer cell ferroptosis	Promote oxidative stress in tumor cells	[113]
MRTX1133	Chemically synthesized drug	In advanced non-small cell lung cancer with KRAS G12D mutation, phase 3	Inhibiting KRAS G12D mutation, eliminating ROS, and alleviating intratumoral immunosuppression	Promote oxidative stress in tumor cells	[173,210]
Lapatinib	Chemically synthesized drug	In advanced or metastatic breast cancer, phase 1	Inhibiting EGFR and apoptotic pathways	Promote oxidative stress in tumor cells	[174,211]

## 10. Conclusions

After reviewing the above information, it can be concluded that the effects of oxidative stress on tumorigenesis and development have three characteristics.

First, tumor development is a multi-stage dynamic process involving at least three defined phases: initiation, promotion, and progression. Oxidative stress is directly involved in all phases of the process and plays a key role (Figure 5). Currently, the biggest obstacle to cancer treatment is the diversity of mechanisms of tumor induction and escape, that is to say, one target gene, pathway, or process is insufficient and a comprehensive target is necessary. Oxidative stress touches the spot.

Second, the processes by which oxidative stress acts on tumor development are crosslinked. The different processes described above are not independent but connected with each other. For instance, tumoral metabolic reprogramming has a close connection with proliferation [212], in which ROS plays a pivotal role via regulating both of them. Furthermore, oxidative stress can affect cellular aging and the composition of the senescence-associated secretory phenotype (SASP) depends on different stimuli, such as pro-inflammatory factors and chemokines, which are associated with inflammatory responses [213], further affecting immunity and cancer. Moreover, oxidative stress is connected with other stresses, such as hypoxia and metabolic stress. It has been observed that different stressors and their responses are crossed. As abscisic acid plays a crucial role in cross-adaptation in plants, ROS could be the hinge of stressors, and some key factors such as FOXO3a and FOXO3a respond to different types of stress through differential modulation [143]. ROS and HIF-1α, which respond to hypoxia inhibition, attenuate each other’s expression [214]. HIF-1a has a comprehensive effect in cancer [215]. Metabolic stress can trigger ROS signaling via the AMPK and AKT pathways [216], and AMPK is closely correlated to carcinogenesis and cancer drug resistance [217].

Last but not least, mitochondria play a crucial role in the impact. (1) As mentioned earlier, mitochondria are the main source and control center of ROS, with both the central enzyme family for ROS production and the majority ROS scavenging systems, such as SOD [218], GSH [219], and MnSOD [220], located there. (2) The target processes of tumor cell metabolic reprogramming, such as the tricarboxylic acid cycle and electron transport chain, are carried out in mitochondria. (3) Signaling molecules associated with tumorigenesis, such as STAT3 [132], TERT [221], PML [222], p53 [223], and HSP60 [213], can also be found in mitochondria. (4) Mitochondria are connected to other subcellular organelles involved in regulating tumorigenesis, such as the nucleus [224] and endoplasmic reticulum (ER). (5) Mitochondria are closely related to the aging of tumor cells. Mitochondrial dysfunction is one of the hallmarks of cell senescence [225] and arouses the mitochondrial dysfunction-associated senescent response (MiDAS) [226]. (6) Mitochondria play a key role in apoptosis and ferroptosis of tumor cells. Mitochondria are the executors of the intrinsic pathway of apoptosis, with apoptosis-related proteins such as the Bcl-2 family proteins and Cyt C [130]. Besides controlling ROS levels, mitochondria also contain large amounts of irons and possess the ability to take up iron. Mitochondrial ferritin is closely associated with oxidative stress and ferroptosis [86,215]. (7) Mitochondria are involved in tumor immunity. It has been demonstrated that the mass, quantity, and morphology of mitochondria can directly regulate immune [227].

Notably, not only is cancer connected with oxidative stress, but a variety of stresses and stress responses, including cellular senescence and cell death, are related to it. There are more factors and pathways than expected involved in cellular senescence, including but not limited to sirtuin, TERT [226], and PML [228]. As illustrated earlier, oxidative stress promotes death escape by way of inducing gene mutation, setting survival signals on, and suppressing immune system function, eventually achieving immortality. Therefore, oxidative stress could be the intersection of these process and play a pivotal role in cell fate determination (Figure 6).

## Figures and Tables

**Figure 1 cells-13-00441-f001:**
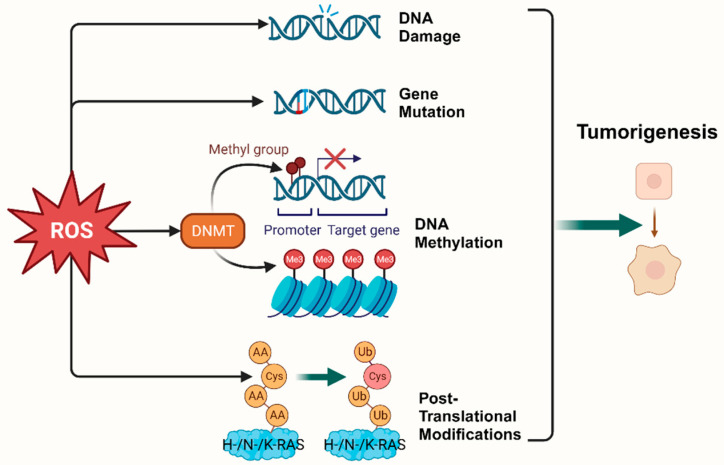
Oxidative stress plays a pivotal role in tumorigenesis. (Created with BioRender.com).

**Figure 2 cells-13-00441-f002:**
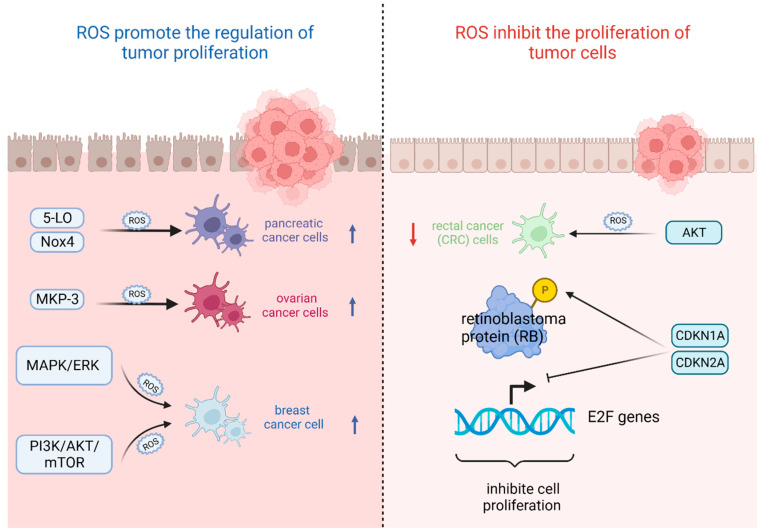
Oxidative stress based on ROS levels will directly affect tumor cell growth state. (Created with BioRender.com).

**Figure 3 cells-13-00441-f003:**
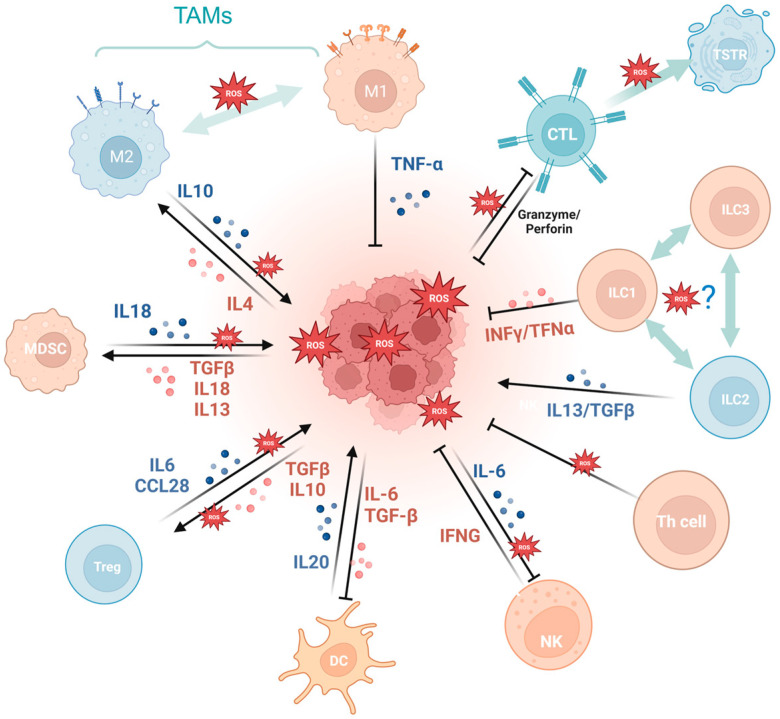
Oxidative stress is involved in tumor immunity. MDSCs, myeloid-derived suppressor cells; Tregs, regulatory T cells; TAMs, tumor-associated macrophages; CTL, cytotoxic T lymphocyte; TSTR, T cell stress response state; ILC, innate lymphoid cell. (Created with BioRender.com).

**Figure 4 cells-13-00441-f004:**
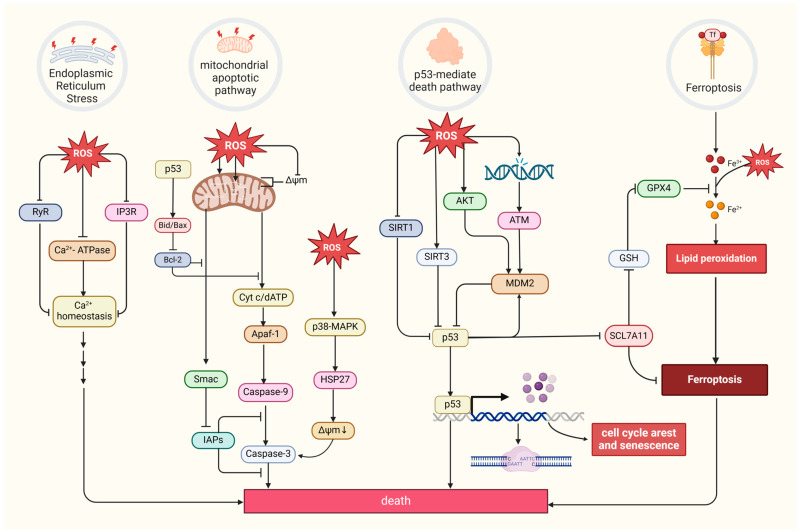
Four primary forms of tumor cell death and regulation of oxidative stress. MDM2, a p53-specific E3 ubiquitin ligase, as a potential target for activating p53 function in cancer therapy, mediates p53 degradation while responding to oxidative stress via being phosphorylated by AKT, ATM, and c-Abl under the state of oxidative stress [145,147]. SIRT1 and SIRT3, as stress responders, deacetylate and inhibit p53, inducing apoptosis in HCC [148,149]. (Created with BioRender.com).

**Figure 5 cells-13-00441-f005:**
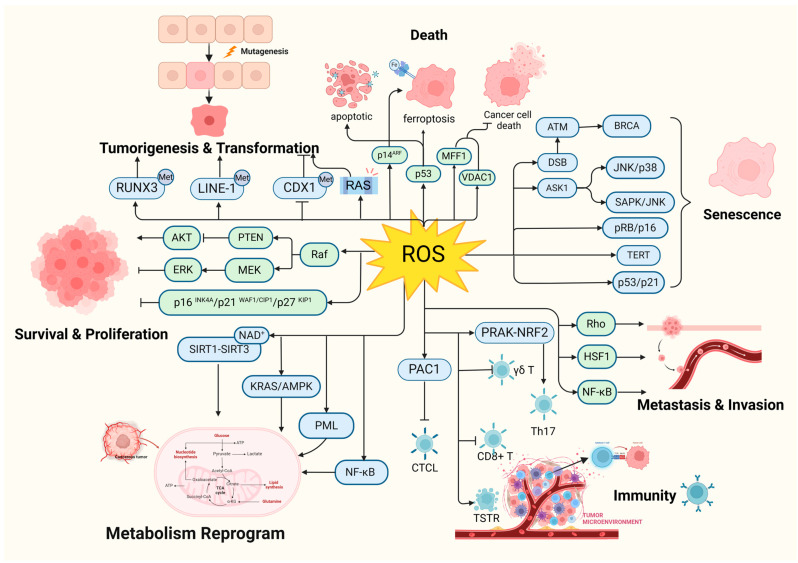
Oxidative stress is involved in all phases of tumorigenesis. RAS: HRAS, NRAS, and KRAS; CTCL: cutaneous T-cell lymphoma; γδ T: γδ T cells. (Created with BioRender.com).

**Figure 6 cells-13-00441-f006:**
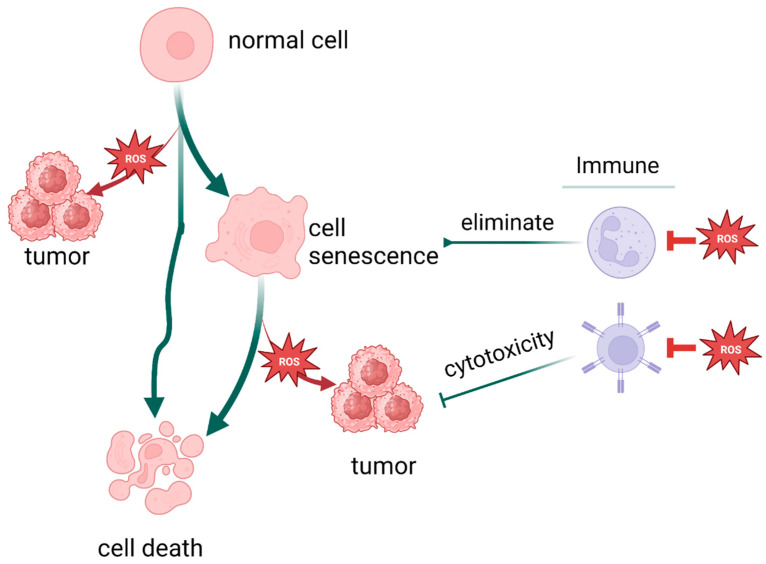
Oxidative stress plays a pivotal role in cell fate determination. (Created with BioRender.com).

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
