# Peer review of "The Role of Oxidative Stress in Tumorigenesis and Progression"

_cells, 2024, doi:10.3390/cells13050441_

Round 1

Reviewer 1 Report (Previous Reviewer 1)

Comments and Suggestions for Authors

The authors have addressed all the reviewer's concerns in a satisfactory manner. The manuscript is now suitable for publication.

Author Response

Dear reviewer 1,

Re:  2844674

Thank you very much for taking your time reviewing this manuscript. I really appreciate all your great comments and suggestions. Please find the revised version of our manuscript entitled " The Role of Oxidative Stress in Tumorigenesis and Progression" which we would like to resubmit for your consideration.

Your comments were highly insightful and greatly improved the quality of our manuscript. We have addressed these constructive comments. The point-by-point responses to the reviewers’ and previous editor comments are listed below.

Major comment 1.  The authors have addressed all the reviewer's concerns in a satisfactory manner. The manuscript is now suitable for publication.

Response: We sincerely thank the reviewer for this recognition.

The revisions in the manuscript are highlighted in red font and yellow highlighted. The entire manuscript has undergone appropriate English editing.

We hope that this version of the manuscript and our accompanying responses will be sufficient to make our manuscript suitable for publication in Cells.

I look forward to meeting your expectations.

With best wishes.

Yours sincerely.

Changshan Wang

Reviewer 2 Report (New Reviewer)

Comments and Suggestions for Authors

Significance of oxidative stress in tumorigenesis, highlighting its involvement in cell senescence, apoptosis, and various diseases including lung cancer, glioma, and leukemia. It explores how oxidative stress affects cell fate determination through processes like mitochondrial stress, endoplasmic reticulum stress, and ferroptosis. The manuscript emphasizes the complex relationship between oxidative stress, cancer progression, and mitochondrial regulation. Understanding these effects is crucial for developing effective tumor therapies. Overall, the manuscript provides valuable insights into the multifaceted role of oxidative stress in tumorigenesis.

Minor Comments:

1. Authors have to discuss the treatment of Receptor Tyrosine Kinase (RAS and EGFR) inhibitors and ROS production in different cancer types.

2. Authors should discuss the role of ROS production in persister cells and their implications in low-fidelity polymerases and cancer.

Author Response

Dear reviewer 2,

Re:  2844674

Thank you very much for taking your time reviewing this manuscript. I really appreciate all your great comments and suggestions. Please find the revised version of our manuscript entitled " The Role of Oxidative Stress in Tumorigenesis and Progression" which we would like to resubmit for your consideration.

Your comments were highly insightful and greatly improved the quality of our manuscript. We have addressed these constructive comments. The point-by-point responses to the reviewers’ and previous editor comments are listed below.

Major comment 1.  Authors have to discuss the treatment of Receptor Tyrosine Kinase (RAS and EGFR) inhibitors and ROS production in different cancer types.

Response: We thank the reviewer for this important comment. Different curative effect of the treatment of RAS and EGFR inhibitors among different cancer types and the relationship with ROS are discussed in the section of Oxidative Stress and Tumor Treatment in lines 363-371. Clinical studies on RAS and EGFR inhibitors applying in cancer therapy were added in table 1.

Extending table 1.

MRTX1133

Chemically synthesized drugs

In Advanced Non-Small Cell Lung Cancer with KRAS G12D Mutation, phase 3

Inhibiting KRAS G12D mutation eliminating ROS and alleviating intratumoral immunosuppression

Promote oxidative stress in tumor cells

[210] [173]

Lapatinib

Chemically synthesized drugs

In Advanced or Metastatic Breast Cancer, phase 1

Inhibiting EGFR and apoptotic pathways

Promote oxidative stress in tumor cells

[174] [211]

Major comment 2. Authors should discuss the role of ROS production in persister cells and their implications in low-fidelity polymerases and cancer.

Response: We appreciate the reviewer pointing this out. We have added this the role of oxidative stress in cancer though implicating low-fidelity polymerases in the section of Oxidative Stress and Tumorigenesis in lines 65-68.

The revisions in the manuscript are highlighted in red font and yellow highlighted. The entire manuscript has undergone appropriate English editing.

We hope that this version of the manuscript and our accompanying responses will be sufficient to make our manuscript suitable for publication in Cells.

I look forward to meeting your expectations.

With best wishes.

Yours sincerely.

Changshan Wang

Reviewer 3 Report (New Reviewer)

Comments and Suggestions for Authors

The manuscript entitled "The Role of Oxidative Stress in Tumorigenesis and Progression" provides an interesting overview of the impact of reactive oxygen species and the phenomenon of oxidative stress on the processes of both carcinogenesis, metabolism and proliferation of cancer cells, as well as the role of oxidative stress in the development of metastases.

The Authors of the manuscript also draw attention to the role of reactive oxygen species and oxidative stress in the course of aging and cancer progression, emphasizing that this role may, as indicated by the research cited in this manuscript, be multidirectional.

In the next part of the manuscript, the Authors present in an interesting and clear way the current clinical application of anticancer therapies based on oxidative stress, including specific examples of the use of anticancer substances.

I agree with the Authors that oxidative stress is involved in every stage of cancer formation and progression and is one of its main causes. Currently, one of the greatest difficulties in the effective treatment of cancer is the multitude of metabolic processes that take part in its pathogenesis.

The Authors present the most important information and correlations in a clear and aesthetic way in the attached tables and figures.

The review is focused, timely interesting and substantially detailed. I recommend publication of the work because, in my opinion, the manuscript is interesting for the Readers.

Author Response

Dear reviewer 3,

Re:  2844674

Thank you very much for taking your time reviewing this manuscript. I really appreciate all your great comments and suggestions. Please find the revised version of our manuscript entitled " The Role of Oxidative Stress in Tumorigenesis and Progression" which we would like to resubmit for your consideration.

Your comments were highly insightful and greatly improved the quality of our manuscript. We have addressed these constructive comments. The point-by-point responses to the reviewers’ and previous editor comments are listed below.

Major comment 1. The review is focused, timely interesting and substantially detailed. I recommend publication of the work because, in my opinion, the manuscript is interesting for the Readers.

Response: It’s our honor to receive such recognition.

The revisions in the manuscript are highlighted in red font and yellow highlighted. The entire manuscript has undergone appropriate English editing.

We hope that this version of the manuscript and our accompanying responses will be sufficient to make our manuscript suitable for publication in Cells.

I look forward to meeting your expectations.

With best wishes.

Yours sincerely.

Changshan Wang

Reviewer 4 Report (New Reviewer)

Comments and Suggestions for Authors

This is a well written review article explaining the role of oxidative stress in tumor cell proliferation and treatment. It is very well established that cancer cells endogenously produce higher amounts of reactive oxygen species (ROS) promoting cell proliferation and genetic instability, a pre-requisite for cancer development and progression. ROS, generated inside the cells as by-product of normal metabolism, could cause tissue injury and DNA damage but in normal cells, a dynamic balance is maintained between ROS level (pro-oxidant) and antioxidant proteins and enzymes. However, with the age this balance shifts toward pro-oxidant resulting in a chronic increase in ROS. Based upon these observations, in the past, several studies have focused on using anti-oxidants for both preventive and therapeutic approaches in the management of several cancers. Authors explained the role of oxidative stress on tumorigenesis by regulating several aspects of tumor metabolism, immunity, and cell death. Authors also discussed the role of oxidative stress in tumor metastasis and therapy. The figures and self-explanatory and add value to the article. However, few very important topics are missing from this article which should be included as there are hundreds of review articles already published on this topic.

1.       Given the role of oxidative stress in promoting several aspects of tumorigenesis, several clinical trials were conducted to test the anti-cancer efficacy of anti-oxidants like SELECT and ATBC trials in Prostate cancer. However, none of the trials showed positive data and failed to achieve the expected goals. Authors should include a section summarizing the clinical trials involving anti-oxidants in several cancers. Authors should also discuss the reasons behind the observed failure.

2.       Several pre-clinical studies in past decade highlighted the potential usage of pro-oxidants in cancer therapy (Few examples are- Nature reviews Drug discovery. 2009;8:579–591. The Journal of biological chemistry. 2008;283:30151–30163. Nat Rev Drug Discov. 2013;12:931–947. Mol Carcinog. 2018;57(1):57–69.). Authors should also include a section discussing the role of pro-oxidants in cancer.

3.       Since, authors included a section on aging and tumor, authors may discuss the role of mitochondrial stress response pathways and dysfunction in aging and cancer.

Comments on the Quality of English Language

No major issues detected.

Author Response

Dear reviewer 4,

Re:  2844674

Thank you very much for taking your time reviewing this manuscript. I really appreciate all your great comments and suggestions. Please find the revised version of our manuscript entitled " The Role of Oxidative Stress in Tumorigenesis and Progression" which we would like to resubmit for your consideration.

Your comments were highly insightful and greatly improved the quality of our manuscript. We have addressed these constructive comments. The point-by-point responses to the reviewers’ and previous editor comments are listed below.

Major comment 1. Authors should include a section summarizing the clinical trials involving anti-oxidants in several cancers. Authors should also discuss the reasons behind the observed failure.

Response: We thank the reviewer for this constructive comment. We have discussed some examples of anti-tumor therapies based on oxidative stress failing to reach the expected effect such as SELECT and ATBC clinical trials and the promising therapies like EGFR and KRAS inhibitors and targeting Nrf2. Reasons may contribute to this have been detailedly analyzed in the section of Oxidative Stress and Tumor Treatment.

Major comment 2. Authors should also include a section discussing the role of pro-oxidants in cancer.

Response: We appreciate the reviewer pointing this out. We have replenishes the role of pro-oxidants in cancer in the section of Oxidative Stress and Tumor Treatment and table 1. Comparisons between pro-oxidants and anti-oxidants are also discussed in lines 352-361.

Major comment 3. Since, authors included a section on aging and tumor, authors may discuss the role of mitochondrial stress response pathways and dysfunction in aging and cancer. 

Response: We appreciate the reviewer for suggesting this. We have supplemented that mitochondria play a pivotal role in oxidative stress and stress induced premature senescence in cancer. In colorectal cancer cells, artesunate treatment induced mitochondrial dysfunction can drastically spur mitochondrial ROS generation, thereby promoting cell senescence, which have become a vital target of cancer therapy.

Artesunate

Natural active substances

In Hepatocellular Carcinoma, phase 1

Promoting the accumulation of intracellular lipid peroxides to induce cancer cell ferroptosis

Promote oxidative stress in tumor cells

[113]

Extending table 1.

The revisions in the manuscript are highlighted in red font and yellow highlighted. The entire manuscript has undergone appropriate English editing.

We hope that this version of the manuscript and our accompanying responses will be sufficient to make our manuscript suitable for publication in Cells.

I look forward to meeting your expectations.

With best wishes.

Yours sincerely.

Changshan Wang

This manuscript is a resubmission of an earlier submission. The following is a list of the peer review reports and author responses from that submission.

Round 1

Reviewer 1 Report

Comments and Suggestions for Authors

Li et al. have submitted a comprehensive literature review manuscript compiling relevant information on oxidative stress implications in cancer.  The article is well organized, and the figures are educative and well-summarize the exposed information. Although the subject of the manuscript may be interesting for the journal readership, some concerns should be addressed by the authors at this stage.

- It is suggested to describe the main features of the ROS molecule members better; the current description is a bit scarce and needs to be more deeply discussed.

-Although the authors try to include several sections to cover the relevant topics, some seem weaker than others according to the discussion and included information. So, sections 3-5 must be improved to balance the importance with other better-addressed sections.

- the title of section 7. Oxidative Stress in The Relationship of Death and Tumor: Since the tumor can be a multicellular entity,  authors should clarify if they refer to cancer/transformed cells or also include other tumor-associated cells. 

-Section 8. “Oxidative Stress and Tumor Treatment,” should be elaborated better since it is key in importance to translational medicine. Regarding this, if there is, can the authors include clinical trials targeting Oxidative Stress in cancer treatment? Also, the clinical perspective of whether to target oxidative stress or promote it to improve cancer therapy is better.

-In the conclusion section, “Scheme 116” is confusing. Can the authors verify this issue?

Comments on the Quality of English Language

Although the English is readable it needs some improvement.  

Reviewer 2 Report

Comments and Suggestions for Authors

The effect of oxidative stress on carcinogenesis is addressed in this review, along with how it affects various stages and cell fate decisions. Whether through mechanisms like mitochondrial stress, endoplasmic reticulum stress, and ferroptosis, it affects carcinogenesis and tumor growth. Senescence, cell death, and cancer all have intricate ties with how oxidative stress affects a cell's destiny. Mitochondria act as a crucial regulatory hub, influencing both the promotion and inhibition of carcinogenesis and tumor growth. New methods for treating cancer can be influenced by a better comprehension of how oxidative stress contributes to carcinogenesis.

This is a well-written manuscript, but it needs some more attention to improve its quality.

Comments-

1-The most prevalent ROS that can affect lipids are hydroxyl (HO•) and hydroperoxyl (HO•2) radical. Lipid peroxidation (LPO), which is mostly localized in the cellular membrane and results in a loss of membrane property, is caused by oxidative damage to lipids. Other molecules may consequently sustain damage from their reactive byproducts. Cell defense mechanisms result in adaptation when exposed to low levels of LPO, however a higher level of LPO causes apoptosis or necrosis. MDA has been the subject of the most research among the numerous distinct aldehydes that can be produced as secondary products during LPO. The author must also discuss how reactive oxygen species harm proteins and lipids in addition to DNA.

2-Since both endogenous and exogenous antioxidants can prevent and repair damage brought on by ROS, the author must also discuss how the altered antioxidant defense system in cancer. In order to strengthen the immune system and reduce the danger of sickness and cancer, they are known as "free radical scavengers." SOD, GPx, and CAT are examples of enzyme-based antioxidants that work by chelating superoxide and other peroxides. They serve as natural antioxidant defense mechanisms, removing ROS production and buildup from cells while preserving redox equilibrium. SOD, which catalyzes the dismutation of superoxide anion radical (O2 •-) into hydrogen peroxide, serves as the first line of defense against free radicals (H2O2).

3-The author must also explain how natural polyphenols' positive effects result from their capacity to scavenge free radicals produced internally, as well as those produced by radiation and xenobiotics. According to some studies, the phenolic compounds' antioxidant qualities may also act as prooxidants to start a ROS-mediated cellular DNA break, which would lead to cell death. 

Comments on the Quality of English Language

minor corrections needed.
